# Adaptable Logical Control for Large Language Models

**Honghua Zhang**[*]
UCLA
hzhang19@cs.ucla.edu

**Po-Nien Kung**[*]
UCLA
ponienkung@cs.ucla.edu

**Masahiro Yoshida**[†]
UCLA & Sony Group Corporation
masahiroyoshida@ucla.edu

**Guy Van den Broeck**
UCLA
guyvdb@cs.ucla.edu

**Nanyun Peng**
UCLA
violetpeng@cs.ucla.edu

## Abstract

Despite the success of Large Language Models (LLMs) on various tasks following human instructions, controlling model generation to follow strict constraints at inference time poses a persistent challenge. In this paper, we introduce Ctrl-G, a neuro-symbolic framework that enables tractable and adaptable control of LLM generation to follow logical constraints reliably. Ctrl-G combines any production-ready LLM with a Hidden Markov Model (HMM), guiding LLM outputs to adhere to logical constraints represented as deterministic finite automata. We show that Ctrl-G, when a TULU2-7B model is coupled with a 2B-parameter HMM, outperforms GPT4 in text editing: on the task of generating text insertions/continuations following logical constraints, our approach achieves over 30% higher satisfaction rate in human evaluation. When applied to medium-size language models (e.g., GPT2-large), Ctrl-G also beats its counterparts on standard benchmarks by large margins. Additionally, as a proof-of-concept study, we use Ctrl-G to assist LLM reasoning on the GSM benchmark, foreshadowing the application of Ctrl-G, as well as other constrained generation approaches, beyond traditional language generation tasks.

## 1 Introduction

Large language models (LLMs) have achieved remarkable performance on a wide range of challenging language generation tasks including translation [4, 48, 41], summarization [49], and open-domain creative generation [45, 38]. Nevertheless, many downstream applications benefit from fine-grained control of LLMs to follow logical constraints, e.g., avoid using bad words for detoxification [9, 1] or inserting text that is coherent with contexts for document revision [16]. Despite the recent advancement of LLM finetuning techniques such as instruction-tuning [5, 42, 35] and preference optimization [28, 33], LLMs still fail to reliably follow logical constraints [37, 20].

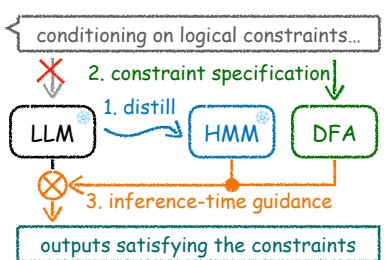

Figure 1: Ctrl-G pipeline; both the LLM and the HMM are frozen once trained.

The major difficulty of achieving constrained generation from LLMs lies in the intractability of conditioning LLMs on logical constraints [34]. One recently proposed framework called GeLaTo [47] uses tractable generative models, which *can* be conditioned on logical constraints efficiently, to guide autoregressive generation from LLMs. Though GeLaTo guarantees that the logical constraints

---

[*]Equal contributions.

[†]Work done at UCLA as a visiting scholar.

38th Conference on Neural Information Processing Systems (NeurIPS 2024).

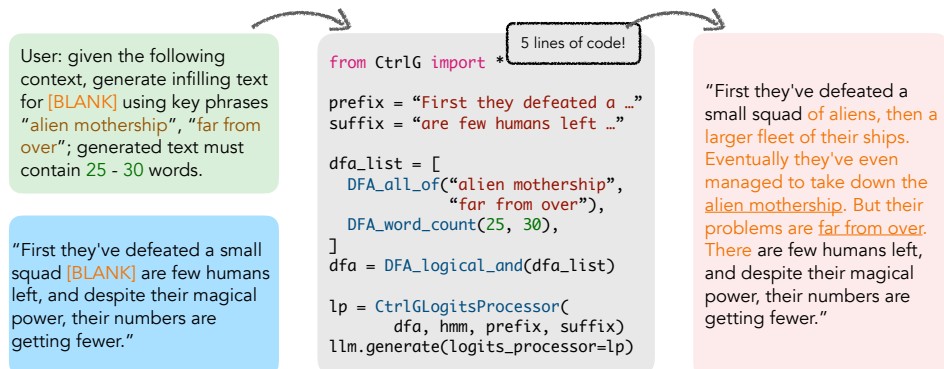

Figure 2: An example usage of Ctrl-G for text insertion with multiple constraints.

will be satisfied, it only works for the keyword-inclusion constraint. Significantly generalizing the GeLaTo framework, we propose Ctrl-G (shorthand for controllable generation while mimicking the keyboard shortcuts Ctrl-C and Ctrl-V) for **reliable**, **scalable** and **adaptable** control of LLMs to follow logical constraints. Ctrl-G consists of three major steps (see Fig. 1): (1) *distillation*: given a LLM, we distill a Hidden Markov Model as its white-box approximation; (2) *constraint specification*: we construct a deterministic finite automaton (DFA) to (compactly) represent the desired logical constraint; (3) *inference*: we condition the HMM on the DFA-specified constraint and compute this conditional probability to steer LLM generation towards satisfying the constraint.

Ctrl-G[3] has three major advantages compared to its counterparts: (1) the desired logical constraints are guaranteed to be satisfied [47]; (2) once we have the distilled HMM, it can be applied to arbitrary constraints without retraining; (3) Ctrl-G works for any constraints specified as DFAs, which can be easily constructed for various applications by leveraging existing algorithms.

We evaluate Ctrl-G on the task of text editing: in the domain of story writing, we evaluate models' ability to generate suggestions for text insertions/continuations under combinations of logical constraints (e.g. keyphrase inclusion and length control; see Fig. 2). Human evaluation shows that Ctrl-G, where a TULU2-7B model [13] is combined with a 2B-parameter HMM, outperforms prominent LLMs including GPT3.5 and GPT4 [27] by over 30% in overall satisfaction rate (i.e., percentage of the generated text that is not only fluent but also satisfies the constraints). We note that as the constraints become more complex, while the generation quality of GPT4 declines, Ctrl-G consistently produces high-quality text, highlighting its strong generalizability to complex constraints. Even when no constraint is present, Ctrl-G still matches with the generation quality of GPT4 in text insertion.

In addition, we demonstrate the extensive adaptability of Ctrl-G on two commonly used benchmarks: commonsense generation [18] and text infilling [7]. When applied to variants of the GPT2 models, Ctrl-G outperforms prior constrained generation approaches by producing outputs of substantially higher quality while achieving 100% constraint satisfaction.

To further explore the potential of Ctrl-G, as a proof-of-concept, we conduct an empirical study on the Grade School Math (GSM) benchmark [6]; here, we use Ctrl-G to assist the LLM reasoning process by enforcing keyphrase-inclusion constraints. Performance improvement suggests the potential of Ctrl-G in applications of a scope broader than the traditional constrained generation tasks.

## 2   Preliminaries

In this section, we briefly summarize the background for (logically-)constrained generation and the basics for Hidden Markov Models. Notations introduced here will be used throughout the paper.

**Constrained generation**   For simplicity, we assume that the lengths of token sequences generated by LLMs are always bounded by some number $n$ and denote the LLM distribution as $p_{\text{lm}}(x_{1:n})$[4]. Given

---

[3]Code available at https://github.com/joshuacnf/Ctrl-G.
[4]Sequences padded to the length of $n$ tokens.

logical constraint $\alpha$, our goal is to generate from $p_{\text{lm}}(x_{1:n} \mid \alpha)$, which decomposes autoregressively:

$$p_{\text{lm}}(x_{1:n} \mid \alpha) = \prod_t p_{\text{lm}}(x_t \mid x_{<t}, \alpha), \quad \text{where} \quad p_{\text{lm}}(x_t \mid x_{<t}, \alpha) \propto p_{\text{lm}}(x_t \mid x_{<t}) \cdot p_{\text{lm}}(\alpha \mid x_t, x_{<t});$$

that is, given that we have generated the first $t-1$ tokens $x_{<t}$, we want to generate the next token $x_t$ from $p_{\text{lm}}(x_t \mid x_{<t}) \cdot p_{\text{lm}}(\alpha \mid x_t, x_{<t})$. The first term $p_{\text{lm}}(x \mid x_{<t})$ is just the next-token distribution of the LLM, but the marginal probability $p_{\text{lm}}(\alpha \mid x_t, x_{<t})$, which characterizes how likely the constraint $\alpha$ will be satisfied *in the future*, cannot be efficiently computed; specifically,

$$p_{\text{lm}}(\alpha \mid x_t, x_{<t}) = \sum_{x_{>t} \text{ s.t. } x_{1:n} \text{ satisfies } \alpha} p(x_{>t} \mid x_t, x_{<t});$$

that is, we need to marginalize over all possible future sequences $x_{>t}$ such that, together with $x_{\le t}$, satisfy $\alpha$. For example, say $\alpha$ is the constraint that the phrase "in the park" must appear at the end of the generated text; to compute the desired marginal probability, we need to enumerate over all future token sequences with this phrase at the end, and there are exponentially many of them.

**Prior work** To solve the problem of constrained generation, one line of work proposes search-based decoding algorithms like NeuroLogic Decoding [22, 21], which explicitly performs heuristic search to find high-probability token sequences that would (partially) satisfy the logical constraint; however such methods scale poorly because the search space grows exponentially with respect to the sequence length. The other line of works including GeDi [15], FUDGE [44] and NADO [25] train auxiliary neural classifiers to approximate the intractable term $p_{\text{lm}}(\alpha \mid x_t, x_{<t})$; however, they do not guarantee that the constraints will be satisfied and the classifiers need to be retrained for different constraints. Some other methods use approximate inference techniques (e.g., sequential Monte Carlo sampling) to approximate the intractable conditional distributions [30, 11, 17], which provide no guarantee on the convergence rate and often suffer from the high-variance of sampling.

**From GeLaTo to Ctrl-G** A recent framework called GeLaTo [47] uses tractable generative models, in particular, Hidden Markov Models (HMMs), to guide LLM generation to satisfy the given logical constraints. Specifically, GeLaTo first (1) distills an HMM $p_{\text{hmm}}(x_{1:n})$ to approximate the LLM distribution $p_{\text{lm}}(x_{1:n})$ and then (2) computes $p_{\text{hmm}}(\alpha|x_t, x_{<t})$ as an approximation for $p_{\text{lm}}(\alpha|x_t, x_{<t})$. Compared to its counterparts, GeLaTo *guarantees* that the constraints will be satisfied. Nevertheless, two major questions remain unanswered, limiting its downstream applications:

- GeLaTo only handles the keyword-inclusion constraint and it is unclear whether $p_{\text{hmm}}(\alpha|x_{t+1}, x_{1:t})$ can be tractably computed for other logical constraints;
- despite the success of GeLaTo on language models at the scale of $0.1$ billion parameters, it is unclear whether the assumption $p_{\text{hmm}}(\alpha \mid x_{\le t}) \approx p_{\text{lm}}(\alpha \mid x_{\le t})$ would still hold for the more recent LLMs (e.g., Llama2), which have over 100 times more parameters.

We propose Ctrl-G as a generalization of GeLaTo and give positive answers to both questions.

**Hidden Markov Models** A Hidden Markov Model (HMM) [32] represents a joint probability distribution over $n$ observed variables $x_{1:n}$ and $n$ hidden variables $z_{1:n}$. Specifically, for language modeling, $x_t$ represents the token at position $t$ and $z_t$ is the corresponding hidden state; $z_t$ takes values in $\{1, 2, \ldots, h\}$, where $h$ is the *number of hidden states*. An HMM models the joint distribution:

$$p(x_{1:n}, z_{1:n}) = p(x_1 \mid z_1) \cdot p(z_1) \cdot \prod_{2 \le t \le n} p(x_t \mid z_t) \cdot p(z_t \mid z_{t-1});$$

in particular, the parameters of an HMM are given by the initial probability $p(z_1)$, the emission matrix $p(x_t|z_t)$ and the transition matrix $p(z_{t+1}|z_t)$; the number of parameters of HMMs grows quadratically with respect to $h$. To perform inference on HMMs efficiently, we leverage the *Markov property*: $p(x_{\ge t} \mid z_t, x_{<t}) = p(x_{\ge t} \mid z_t)$. For example, we can efficiently compute $p(x_{\le t}) = \sum_{z_t} p(x_{\le t}, z_t)$ by the following recurrence relation, referred to as the *forward algorithm* [32]:

$$p(x_{\le t}, z_t) = \sum_{1 \le z_{t-1} \le h} p(x_t \mid z_t) \cdot p(z_t \mid z_{t-1}) \cdot p(x_{\le t-1}, z_{t-1}).$$

## 3 Tractable probabilistic reasoning over logical constraints

The Ctrl-G pipeline consists of three steps (Fig. 1): (1) *distillation*: we train an HMM on samples drawn from the LLM to minimize their KL-divergence; (2) *constraint specification*: we construct a

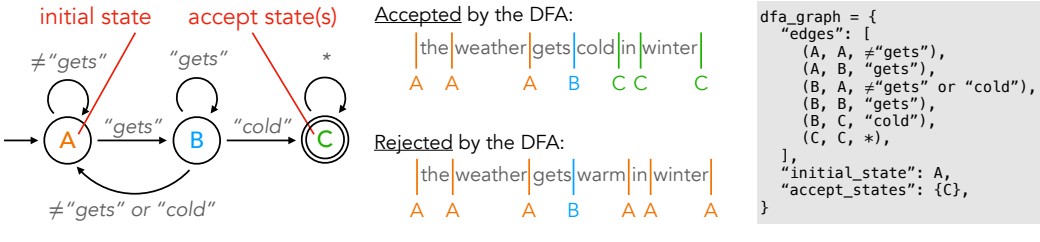

(a) DFA as graph.   (b) Examples of DFA state transition.  (c) Specifying a DFA for Ctrl-G

Figure 3: Example of a DFA representing the logical constraint that the phrase "gets cold" must appear in the generated text along with pseudo-code for representing this DFA in Ctrl-G.

(compact) deterministic finite automaton (DFA) $\mathcal{M}$ representing the desired logical constraint $\alpha$ (i.e., $\mathcal{M}$ accepts $x_{1:n}$ if and only if $x_{1:n}$ satisfies $\alpha$); (3) *inference*: for each step of the autoregressive generation from the LLM, we compute $p_{\text{hmm}}(\alpha \mid x_t, x_{<t})$ as an approximation for $p_{\text{lm}}(\alpha \mid x_t, x_{<t})$ and then sample the next token from

$$p_{\text{ctrl-g}}(x_t \mid x_{<t}, \alpha) \propto p_{\text{lm}}(x_t \mid x_{<t}) \cdot p_{\text{hmm}}(\alpha \mid x_t, x_{<t}); \tag{1}$$

here, given that $\alpha$ is represented as $\mathcal{M}$,

$$p_{\text{hmm}}(\alpha \mid x_t, x_{<t}) = \sum\nolimits_{x_{>t} \text{ s.t. } \mathcal{M} \text{ accepts } x_{1:n}} p_{\text{hmm}}(x_{>t} \mid x_t, x_{<t}) \tag{2}$$

For step (1) (distillation) we follow the procedure proposed by [47], and we describe step (2) and step (3) in Sec. 3.1 and Sec. 3.2, respectively. In the end of this section, we briefly discuss the distinction between pure logical reasoning and probabilistic reasoning over constraints.

## 3.1 Logical constraints as DFAs

Deterministic finite automata (DFAs) [24, 31, 10] are computation models that *accept* or *reject* some given strings. Figure 3a shows an example DFA encoding the constraint that the phrase "gets cold" must appear: it accepts all strings containing this phrase and rejects the others. The DFA consists of 3 different *states* labeled $A$, $B$ and $C$, where $A$ is the *initial state* and $C$ an *accept state*. The states are connected by edges marked with sets of words (tokens, to be precise), which fully specify the *transition function* of the DFA. A DFA decides whether a given string satisfies the constraint by consuming it left-to-right while transitioning from state to state accordingly; in the end, the DFA accepts the string if it is in an accept state. See Figure 3b for an example.

**Definition 3.1.** A *deterministic finite automaton* (DFA) is a tuple $\mathcal{M} = (Q, \Sigma, \delta, q_0, F)$, where $Q$ is a finite set of *states*, $\Sigma$ a finite set of *symbols* (i.e., tokens of an LLM), $\delta : Q \times \Sigma \to Q$ a *transition function*, $q_0$ an *initial state*, and $F \subseteq Q$ a set of *accept states*. A string of tokens $w_1 w_2 \ldots w_n$ is accepted by $\mathcal{M}$ if there exists a sequence of states $q_0, q_1 \ldots q_n$ s.t. $\delta(q_i, w_{i+1}) = q_{i+1}$ for $1 \le i \le n, q_n \in F$.

One question naturally arises: how can we come up with DFA representations for logical constraints? We first note that in the real world, we can always assume that the lengths of the generated token sequences are *bounded by a constant*; hence DFAs can represent any logical constraints defined on this bounded set and the important question is whether we can do this *efficiently*. For many common logical constraints, we can efficiently construct their DFA representations via existing algorithms. For example, given a string consisting of $n$ tokens, to encode the constraint that the string must appear, we can construct a DFA of size $O(n)$ by adapting the well-known *Knuth–Morris–Pratt* (KMP) algorithm [14] for string matching (e.g., Fig. 3a). One can also easily specify *compositional* logical constraints via DFAs by taking their intersection (logical and), union (logical or), complement (logical negation) or concatenation, which we illustrate throughout the rest of this paper.

## 3.2 An efficient algorithm for marginalizing HMMs over DFAs

Now assume that we have a constraint $\alpha$ encoded as a DFA $\mathcal{M}$ with $k$ states $Q = \{1, 2, \cdots k\}$ and $m$ edges, and we are given a distilled HMM with $h$ hidden states. To sample the next token from Eq. 1, we need to compute $p_{\text{hmm}}(\alpha \mid x_t, x_{<t})$, which is the marginal probability over all strings accepted by $\mathcal{M}$ (see Eq. 2). In the following, we describe a tractable algorithm for computing this probability.

In autoregressive generation, $\mathcal{M}$ starts from the initial state and transitions according to the transition function as each new token is generated; we denote the state of $\mathcal{M}$ after sampling the first $t$ tokens $x_{\leq t}$ as $s_t$. In addition, we use the uppercase $S_t$ to denote the *random variable* representing the state of $\mathcal{M}$ after sampling the first $t$ tokens: e.g., $S_n \in F$ denotes the event that the token sequence $x_{1:n}$ is accepted by $\mathcal{M}$. Dropping the subscript "hmm" from $p_{\text{hmm}}(\alpha \mid x_t, x_{<t})$, we compute

$$p(\alpha \mid x_t, x_{<t}) = p(S_n \in F \mid x_t, x_{<t}) = p(S_n \in F, x_t, x_{<t})/p(x_t, x_{<t}).$$

The denominator $p(x_t, x_{<t})$ can be easily computed by the forward algorithm [32]; so we compute

$$
\begin{aligned}
p(S_n \in F, x_t, x_{<t}) &= \sum_{z_t} p(S_n \in F \mid z_t, x_t, x_{<t}) \cdot p(z_t, x_t, x_{<t}) \\
&= \sum_{z_t} \boxed{p(S_n \in F \mid z_t, s_t)} \cdot p(z_t, x_t, x_{<t})
\end{aligned}
\tag{3}
$$

the first step follows from the law of total probability and the second step follows from the Markov properties of HMMs and DFAs, as well as the fact that $s_t$ is fully determined by $x_{\leq t}$. Again, the term $p(z_t, x_t, x_{<t})$ can be computed by the forward algorithm and we reduce the problem to computing the boxed term. We compute $p(S_n \in F \mid z_t, s_t)$ for all $1 \leq t \leq n$, $1 \leq z_t \leq h$ and $1 \leq s_t \leq k$ via the following recurrence relation:

$$\boxed{p(S_n \in F \mid z_t, s_t)} = \sum_{z_{t+1}} p(z_{t+1} \mid z_t) \cdot \sum_{s_{t+1}} \boxed{p(S_n \in F \mid z_{t+1}, s_{t+1})} \cdot \sum_{x_{t+1} \in \text{edge}(s_t, s_{t+1})} p(x_{t+1} \mid z_{t+1}); \tag{4}$$

here $\text{edge}(s_t, s_{t+1}) := \{w : \delta(s_t, w) = s_{t+1}\}$ denotes the set of tokens $w$ that transition $\mathcal{M}$ from $s_t$ to $s_{t+1}$. The base case of the recurrence relation is given by $p(S_n \in F \mid z_n, s_n) = 1$ if $s_n \in F$ and $0$ otherwise. We refer readers to the appendix for its derivation. Algorithm 1 shows the pseudo-code for sampling from $p_{\text{ctrl-g}}(x_{1:n} \mid \alpha)$ autoregressively, using the recurrence relations above.

*Runtime analysis of Algorithm 1.* To sample from Ctrl-G, the computation overhead (i.e. in addition to the LLM inference cost) is dominated by the computation of $p(S_n \in F \mid z_t, s_t)$ for all $t$, $z_t$ and $s_t$ as shown in Eq. 4. Since $\sum_{x_{t+1} \in \text{edge}(s_t, s_{t+1})} p(x_{t+1} \mid z_{t+1})$ does not depend on t, we can precompute and cache their values, resulting a one-time cost of $O(mh|\Sigma|)$. Then, note that for $s_t$, we only need to consider the $s_{t+1}$ where $\text{edge}(s_t, s_{t+1}) \neq \emptyset$. Hence, fixing $t$ and $z_t$, when we compute $p(S_n \in F \mid z_t, s_t)$ for all $1 \leq s_t \leq k$, we only need to (1) enumerate through $1 \leq z_{t+1} \leq h$ and (2) for each $z_{t+1}$, we only need to go through each edge exactly once. There are $m$ edges in total, so it follows that the cost is $O(n \cdot h \cdot h \cdot m) = O(nmh^2)$. The total time complexity is $O(nmh^2 + mh|\Sigma|)$, which simplifies to $O(nmh^2)$ given that $|\Sigma| < nh$ in practice. $\square$

**Theorem 3.2.** *Given a constraint $\alpha$ represented as a DFA with $m$ edges and an HMM with $h$ hidden states, the time complexity for sampling a sequence of $n$ tokens from $p_{\text{ctrl-g}}(x_{1:n} \mid \alpha)$ is $O(nmh^2)$.*

### 3.3 Logical reasoning vs. probabilistic reasoning

Some recent work as well as open source projects have proposed to use regular expressions (regex) to achieve structured generation from LLMs [23, 43, 50]. Regex are equivalent to DFAs in terms of the logical constraints they can represent, but the aforementioned approaches only perform pure *logical reasoning* over regex, which is not suitable for many constrained generation tasks. For example, consider the task of generating a sentence that ends with the phrase " in the park":

- **guidance** [23] (logical reasoning): *silhouette of suspected ... an heavily secured.in the park*

- **Ctrl-G** (probabilistic reasoning): *A man and a woman are walking in the park*

---

**Algorithm 1:** Ctrl-G: sampling $n$ tokens

**Input:** DFA $\mathcal{M} = (Q, \Sigma, \delta, q_0, F)$
       HMM $q_1$, LLM $q_2$.
**for** $t$ **from** $n$ **to** $1$ **do**
    pre-compute $q_1(\alpha \mid z_t, s_t)$ by Eq. 4.
**end for**
**initialize** $s_0 := q_0$, $x_{1:0} := \emptyset$
**for** $t$ **from** $1$ **to** $n$ **do**
    compute $q_1(\alpha \mid x_{<t}, x_t)$ by Eq. 3.
    sample $x_t \propto q_1(\alpha \mid x_{<t}, x_t) \cdot q_2(x_t \mid x_{<t})$
    update $x_{\leq t} := x_{<t} \oplus x_t$
    transition $\mathcal{M}$ from $s_{t-1}$ to $s_t := \delta(s_{t-1}, x_t)$
**end for**
**return** $x_{1:n}$

---

Table 1: CommonGen results. All methods are applied to the GPT2-large model.

| | BLEU-4 | | ROUGE-L | | CIDEr | | SPICE | | Constraint | |
|---|---|---|---|---|---|---|---|---|---|---|
| | dev | test | dev | test | dev | test | dev | test | dev | test |
| *supervised* - base models trained with full supervision | | | | | | | | | | |
| FUDGE | - | 24.6 | - | 40.4 | - | - | - | - | - | 47.0% |
| A*esque | - | 28.2 | - | 43.4 | - | 15.2 | - | 30.8 | - | 98.8% |
| NADO | 30.8 | - | 44.4 | - | 16.1 | - | 32.0 | - | 88.8% | - |
| GeLaTo | 34.0 | 34.1 | 46.2 | 45.9 | 17.2 | 17.5 | 32.2 | **33.5** | **100.0%** | **100.0%** |
| Ctrl-G | **35.1** | **34.4** | **46.7** | **46.4** | **17.4** | **17.6** | **32.7** | 33.3 | **100.0%** | **100.0%** |
| *unsupervised* - base models not trained with keywords as supervision | | | | | | | | | | |
| A*esque | - | 28.6 | - | 44.3 | - | 15.6 | - | 29.6 | - | - |
| NADO | 26.2 | - | - | - | - | - | - | - | - | - |
| GeLaTo | 30.3 | 29.0 | 44.3 | 43.8 | 15.6 | 15.5 | 30.2 | 30.3 | **100.0%** | **100.0%** |
| Ctrl-G | **32.1** | **31.5** | **45.2** | **44.8** | **16.0** | **16.2** | **30.8** | **31.2** | **100.0%** | **100.0%** |

Even though both generations end with " in the park", it is clear that the output from guidance is not desirable as it forcefully appends the phrase to some irrelevant text. The reason is that guidance, by performing pure logical reasoning, only discard the next tokens $x_t$ that would make $\alpha$ unsatisfiable, while the probabilities of the other next tokens remain unchanged; in contrast, Ctrl-G performs *probabilistic reasoning* by estimating $p_{lm}(\alpha \mid x_t, x_{<t})$, i.e., we estimate how likely each next token $x_t$ would eventually lead to $\alpha$ being satisfied. Ctrl-G subsumes the other approaches in the sense that if we set $p_{hmm}(\alpha \mid x_t, x_{<t}) = 1$ for all non-zero values, that is, if we remove all probabilistic information, then it degenerates to pure logical reasoning.

# 4 Evaluating Ctrl-G on constrained generation benchmarks

## 4.1 Commonsense Generation

Following prior work [21, 25], we first evaluate Ctrl-G on the Commonsene Generation (Common-Gen) benchmark [18]. Each test example of CommonGen provides 3 to 5 concepts (keywords) as input and the goal is to generate a natural sentence that incorporates all keywords, allowing for any of their inflections. For example, given *"car"*, *"snow"* and *"drive"* as concepts, both *"a man drives a car on a snow covered road"* and *"the car drove through the snow"* are considered acceptable.

**DFA construction** For CommonGen, given one keyword, say, "snow", we adapt the *Aho-Corasick algorithm* [2] to construct a DFA enforcing the constraint that *at least one of* its inflections (e.g., "snow", "snowing" or "snowy") must appear. To encode the constraint that *multiple* keywords must appear, we can simply take the intersection of the individual DFAs [10]; see appendix for an example.

**Experiments & results** We use the GPT2-large checkpoint (only finetuned for domain adaptation) released by [47] as our base model and we follow the same pipeline to distill an HMM with 32768 hidden states: we sample 4M examples from the base model and train the HMM for 40 EM steps, each consisting of 100K examples. We compare Ctrl-G against FUDGE [44], NADO [25], NeuroLogic A*esque decoding [21] and GeLaTo [47]; GeLaTo uses the same base model as Ctrl-G. The results are summarized in Table 1, where the *Constraint* column shows the percentage of the outputs containing all concepts. Compared to all baselines, Ctrl-G achieves not only 100% constraint satisfaction rate but also substantially higher generation quality measured by automatic evaluation metrics [29, 19, 40, 3].

**Runtime comparison** From an algorithmic perspective, GeLaTo only handles keyword constraints hence it is a special case of Ctrl-G. Nevertheless, Ctrl-G also runs significantly faster than GeLaTo, as shown in Table 2. The GeLaTo implementation only tensorizes the HMM inference component, while the component that reasons about the constraints runs sequentially on CPU. In contrast, by representing DFAs as (weighted) adjacency matrices, Ctrl-G tensorizes the inference procedure for both HMMs and DFAs and runs on GPUs with full parallelization. Besides, both GeLaTo and Ctrl-G runs significantly faster than A*esque, which explicitly performs heuristic search.

**Generalization to more keywords** To evaluate the generalization performance of Ctrl-G, we construct test examples containing 6 to 9 concepts (CommonGen+): we randomly select 100 examples with 5 concepts from the dev split of CommonGen, and then augment them with additional keywords

Table 2: Time (seconds) of generating one example on CommonGen (dev); # of HMM hidden states shown in brackets. Beam sizes used by A*esque, GeLaTo and Ctrl-G are 20, 128 and 128.

| | | unsupervised | | | supervised | |
|---|---|---|---|---|---|---|
| # of concepts | 3 | 4 | 5 | 3 | 4 | 5 |
| A*esque | 472.9 | 542.5 | 613.9 | 8.5 | 9.6 | 11.4 |
| GeLaTo [4096] | 69.8±32.3 | 97.9±39.5 | 143.0±44.4 | 49.8±20.8 | 88.7±30.5 | 127.6±30.4 |
| Ctrl-G [4096] | 1.1±0.3 | 1.9±0.5 | 4.6±1.4 | 1.2±0.4 | 2.3±0.8 | 5.7±1.7 |
| Ctrl-G [32768] | 4.1±0.9 | 9.0±2.0 | 22.3±5.4 | 4.7±1.6 | 11.0±3.8 | 27.6±8.3 |

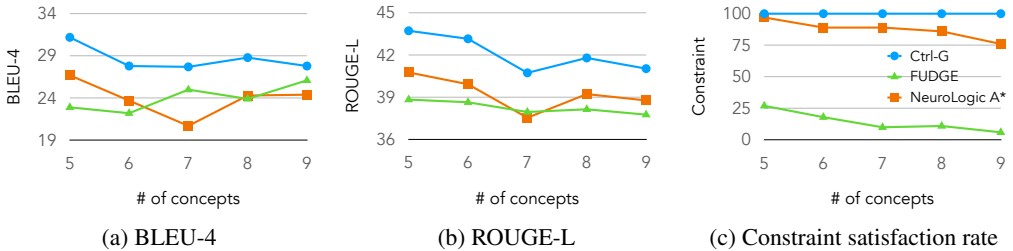

(a) BLEU-4      (b) ROUGE-L      (c) Constraint satisfaction rate

Figure 4: CommonGen+ results; Ctrl-G generalizes well on test examples with more than 5 concepts.

sampled from their reference sentences. As shown in Fig. 4, Ctrl-G achieves 100% constraint satisfaction rate while preserving high generation quality across all settings.

## 4.2 Text infilling

We also evaluate Ctrl-G on a text infilling benchmark [7] constructed from the ROC stories corpus [26]. Each test example consists of a short story with some fragments masked out, each of a specified granularity; the goal is to fill in the masks. Here is an example: *"Jill wanted to knit her [WORD] a sweater. [SENTENCE] She finished [NGRAM] for her boyfriend's birthday. Jill was [WORD]."*

**DFA construction** The underlying logical constraint for the task of text infilling is similar to that of CommonGen. We can view the non-masked parts, e.g., "Jill wanted to knit her" and "a sweater." from the example above, as keyphrases, and the task reduces to generating a piece of text such that all keyphrases appear in the given order. In this setting, given $k$ text fragments, we first construct $\mathcal{M}_1, \ldots, \mathcal{M}_k$ using the KMP algorithm [14]; then, we *concatenate* them to represent the constraint that they must appear in the given order. Though DFA concatenation is intractable in general [46], we observe that the KMP DFAs can actually be concatenated in linear time. See appendix for details.

Table 3: Text infilling results (BLEU-4/ROUGE-L) across different masking ratios.

| | | BLEU-4 | | | | ROUGE-L | | |
|---|---|---|---|---|---|---|---|---|
| mask ratio | 13% | 21% | 32% | 40% | 13% | 21% | 32% | 40% |
| ILM | **85.2**±0.1 | 76.3±0.1 | 64.3±0.1 | 53.8±0.1 | **90.9**±0.2 | **84.9**±0.3 | 76.3±0.4 | 68.4±0.5 |
| Ctrl-G | **85.4** | **77.5** | **66.5** | **57.2** | 90.6 | **85.2** | **77.0** | **69.8** |
| diff. | +0.2 | +1.2 | +2.2 | +3.4 | −0.3 | +0.3 | +0.7 | +1.4 |

**Experiments & results** We use the GPT2-small checkpoint (only finetuned for domain adaptation with no supervision on the task of text infilling) released by [7] as the base model for Ctrl-G and compare against the ILM model, which is a GPT2-small model trained on this text infilling benchmark with full supervision. By applying the mask function from [7], we construct 4 test sets with different masking ratios (i.e., different percentage of masked characters) by changing the hyper-parameters. We measure the BLEU and ROUGE scores of the completed stories (i.e., including both the masked and unmasked parts) with respect to the original stories. The ILM model adopts sampling for decoding, so we run the ILM inference for 10 times to report the means and standard deviations. The results are summarized in Table 3. Based on [7], ILM is trained on a distribution with a masking ratio of approximately 15%, explaining why it achieves the best performance on the test set with 13% masking

ratio. Note that the performance gap between Ctrl-G and ILM improves almost monotonically as the masking ratio increases, again illustrating the strong generalization performance of Ctrl-G.

# 5 Scaling up Ctrl-G for interactive text editing

Human-AI collaborative writing has been a long studied topic in the Human-Computer Interaction (HCI) community [12, 36]. One prior work [16] proposed CoAuthor, a graphical user interface for querying LLMs to generate continuation/insertion suggestions in arbitrary positions of a document. However, when using CoAuthor to ask for LLM suggestions, users are unable to specify their preferences. We propose to extend the CoAuthor system by allowing users to have fine-grained control over the suggestions generated by LLMs: for example, users can control the topic of the generated content by instructing LLMs to incorporate certain keyphrases, and they can also ask for more concise/detailed suggestions by controlling their lengths. For this application, we apply Ctrl-G to the TULU2-7B model and compare against prominent LLMs including GPT3.5 and GPT4.

## 5.1 Experiment setup

**Dataset construction**   We construct an evaluation dataset consisting of 800 test examples, each based on one story passage extracted from the CoAuthor dataset [16]. These stories are jointly written by humans and the GPT3.5-turbo-instruct model, falling under ten different topics. For each story, we randomly split it into *prefix*, *infix* and *suffix*; we mask out the *infix* and view it as a gold reference. We consider two scenarios when evaluating the models: **continuation** and **insertion**. For continuation, we only provide *prefix* to the model, and the model is supposed to generate one suggestion for continuation; for insertion, we provide both *prefix* and *suffix* to the model and the model is required to generate a piece of text that is coherent with both *prefix* and *suffix*. Additionally, we consider imposing combinations of the following two constraints:

- **Keyphrase**: suggestions should include one to three given keyphrases.
- **Word Count**: suggestions should contain $a$ to $b$ words where $1 \leq a \leq b \leq 32$.

We consider all combinations of the following settings: insertion or continuation, w/ or w/o keyphrase constraint, w/ or w/o word-count constraint, resulting in 8 different settings. For each setting, we sample 100 stories from the CoAuthor dataset and create 100 test examples (e.g., Fig. 2).

**Scaling up Ctrl-G**   We adopt the TULU2-7B [13] model, which is an instruction-tuned variant of the Llama2 [39] model with 7 billion parameters, as the base model for Ctrl-G. We further finetune the base model on 3000 examples extracted from the WritingPrompt dataset [8] for the task of text continuation, following the prompt "Continue the given text:" along with a story prefix. After finetuning, we use the same prompt to sample 5 million examples from the base model and train an HMM with 32768 hidden states (approx. 2 billion parameters). Note that for the task of text insertion, the base model *only sees the prefix*, while the suffix is incorporated as a part of the constraint $\alpha$; i.e., the HMM is fully responsible for guiding the base model to generate a piece of text that will be coherent with the suffix. For generation, we sample 128 examples from $p_{\text{ctrl-g}}$ with temperature $0.7$ and pick the one with the highest likelihood given by the base model as the final output.

**Baselines**   We compare Ctrl-G against prominent LLMs including the GPT3.5 model and the GPT4 model. To generate output from the GPT models, we adopt the prompt provided by the OpenAI documentation for text insertion/continuation, with constraints specified in the instructions. See appendix for the specific prompt templates. In addition to the GPT models, we also compare Ctrl-G against pure instruction-tuning: specifically, we construct 1000 training examples for the task of text insertion based on the WritingPrompt dataset and further finetune the TULU2-7B model for text insertion, following the prompt *"Generate the text at [INSERT_TEXT] tag:\n{prefix}[INSERT_TEXT]{suffix}."* For all baselines, for the purpose of fair comparison, we generate 128 samples for each test example and select the one with the highest probability as the final output.

**Human evaluation**   To evaluate the quality of the generated outputs, we conduct human evaluation through the Amazon Mechanical Turk (MTurk) platform. For each test example, we generate the outputs from TULU2 (prompt only), GPT3.5, GPT4 and Ctrl-G respectively, and ask annotators to rate their quality on a scale from 1 to 5. For each test example, we present the generated outputs from all models, along with their original context, to the annotators side-by-side and ask them to evaluate their quality; specifically, we ask the annotators to answer the following questions:

Table 4: Evaluation results of interactive text editing. *K&W* indicates that the model should adhere to both keyphrase (*K*) and word count (*W*) constraints simultaneously. We present the human evaluation score (*Quality*), constraint success rate (*Success*), and overall satisfaction rate (*Overall*), which represents the proportion of examples meeting logical constraints with a Quality score above 3.

| | Continuation | | | | | Insertion | | | | |
|---|---|---|---|---|---|---|---|---|---|---|
| | *None* | *K* | *W* | *K&W* | *Avg.* | *None* | *K* | *W* | *K&W* | *Avg.* |
| *Quality* | | | | | | | | | | |
| TULU2 | 3.80 | 3.77 | 3.87 | 3.88 | 3.83 | 2.68 | 2.64 | 2.78 | 2.74 | 2.71 |
| GPT3.5 | 4.40 | 4.32 | **4.44** | **4.36** | 4.38 | 2.27 | 2.22 | 2.27 | 2.31 | 2.27 |
| GPT4 | **4.48** | **4.44** | **4.44** | 4.26 | **4.40** | **3.79** | 3.33 | 3.53 | 3.10 | 3.44 |
| Ctrl-G | 4.13 | 3.98 | 4.27 | 3.96 | 4.08 | **3.77** | **3.56** | **3.73** | **3.59** | **3.67** |
| *Success* | | | | | | | | | | |
| TULU2 | - | 35% | 33% | 1% | 23% | - | 12% | 20% | 3% | 12% |
| GPT3.5 | - | 36% | 62% | 31% | 43% | - | 22% | 54% | 10% | 29% |
| GPT4 | - | 56% | 55% | 59% | 57% | - | 60% | 20% | 27% | 36% |
| Ctrl-G | - | **100%** | **100%** | **100%** | **100%** | - | **100%** | **100%** | **100%** | **100%** |
| *Overall* | | | | | | | | | | |
| TULU2 | - | 30% | 31% | 1% | 21% | - | 7% | 10% | 1% | 6% |
| GPT3.5 | - | 36% | 62% | 31% | 43% | - | 0% | 5% | 2% | 2% |
| GPT4 | - | 56% | 55% | 57% | 56% | - | 41% | 17% | 14% | 24% |
| Ctrl-G | - | **89%** | **97%** | **90%** | **92%** | - | **76%** | **78%** | **82%** | **79%** |

- *Q1. is the paragraph coherent and grammatically correct?*
- *Q2. is the paragraph consistent and semantically reasonable?*
- *Q3. based on your answers to Q1&Q2, what is your rating for the overall quality?*

Note that we only ask human annotators to evaluate the coherency and fluency of the generated text and they are not aware of the required logical constraints. We ask three annotators to evaluate each output and compute their inter-annotator agreement score. See appendix for more details.

## 5.2 Results

The evaluation results are summarized in Table 4, showing the quality score,[5] constraint satisfaction rate, and overall satisfaction rate. In particular, the overall satisfaction rate denotes the percentage of test examples that (1) satisfy the constraint and (2) attain average quality scores $> 3$. For continuation, in terms of generation quality, GPT4 beats all other models; this is no surprise, as gigantic models like GPT3.5 (with 175B parameters) and GPT4 have significant advantage in generating high quality text continuations. However, despite the high generation quality, the success rates for GPT3.5 and GPT4 are relatively low (the highest 59%) while Ctrl-G always satisfy the specified constraints; hence in terms of the overall satisfaction rate, Ctrl-G beats all baselines by large margins when constraints are present. For the case of insertion, the "implicit" soft constraint here is that the generated parts need to be coherent with the given suffix, which is challenging for autoregressive models; in this case, in terms of pure generation quality, Ctrl-G beats/matches with the performance of GPT4 in all settings; for insertion, the success rate of all baselines becomes even lower compared to continuation, while Ctrl-G achieves 100% success rate in all settings. In terms of overall satisfaction rate, Ctrl-G again beats all baselines. The other observation is that the generation quality of GPT4 decreases as the logical constraints become more complex, while the generation quality of Ctrl-G stays relatively consistent across all settings, demonstrating strong generalization performance.

## 5.3 Runtime analysis

We provide an empirical analysis on the runtime of Ctrl-G, with TULU2-7B as the base model. In addition to the computation cost of the base LLM, the major cost of Ctrl-G lies in the computation of $p_{\text{hmm}}(\alpha \mid x \leq t)$, with a time complexity of $O(nmh^2)$ (Thm. 3.2); here $n$ is the maximum sequence length, $m$ is the size (i.e. # of edges) of the DFA, and $h$ is the number of HMM hidden states. First, fixing the sequence length $n$, we change the size of the DFA and verify that the time for generating each token scales roughly linearly with respect to the DFA size (Fig. 5 left). Then, fixing a DFA of

---

[5]average ratings given to *Q3* in human evaluation; see appendix for complete results.

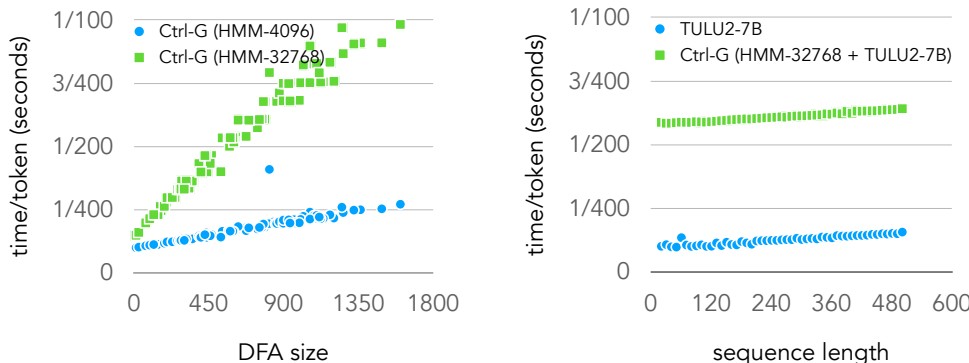

Figure 5: Runtime analysis of Ctrl-G; Left: the generation time per token scales linearly w/ respect to DFA size. Right: the generation time per token stays constant w/ respect to sequence length.

size $\approx 900$, we change the sequence length $n$ and measure the time for generating each token from Ctrl-G and the base LLM respectively. The gap between the two lines in Fig. 5 (right) shows the computation overhead introduced by Ctrl-G, which stays *constant* with respect to the sequence length. On the other hand, however, due to the attention mechanism, the time for generating each token from the base LLM scales linearly with respect to $n$. Hence, the computation cost will be dominated by the base model when generating long sequences. The runtime measurements are conducted on an NVIDIA-A100 GPU with 80GB memory.

## 6    Perspectives: improving LLM reasoning abilities via logical constraints

In this section, we explore the use of Ctrl-G on a non-traditional constrained generation application. As a case study, we apply Ctrl-G to assist the reasoning process of the TULU2-7B model on the grade school math (GSM) benchmark. As we naively apply chain-of-thought prompting, we observe that for 293 out of the 1319 test examples, the model fails to use all numbers provided in the problem statement; this leads to a much lower accuracy on the 293 examples compare to that on the complete test set. For such 293 test examples, we apply Ctrl-G to the TULU2-7B model to enforce the constraint that all numbers from the problem statement must be generated as part of the chain-of-thought reasoning process. We sample 16 outputs from the TULU2-7B model and do a majority vote; with Ctrl-G, the model achieves 28.3% accuracy, which is 3.4% higher than the marjoity-vote accuracy without Ctrl-G.

Our proof-of-concept study on the GSM benchmark illustrates one potential use case of Ctrl-G beyond traditional language generation tasks. Specifically, we demonstrate the possibility of "approximating" soft control (i.e., better reasoning ability in this setting) via logical constraints. For future work, we motivate the application of Ctrl-G, as well as other constrained generation approaches, on a broader scope of downstream tasks: e.g., helping LLM detoxification by conditioning on a set of bad words/phrases not appearing, improving the reasoning ability of LLMs by conditioning on generating longer reasoning sequences, and controlling the topic of the generated content by conditioning on the occurrence of certain keyphrases.

## 7    Conclusion

We propose Ctrl-G, a versatile framework that enables reliable and flexible inference-time control of LLMs; given any production-ready LLM, Ctrl-G distills an HMM as its approximation and uses it to guide the LLM to generate outputs that comply with any logical constraints specified as DFAs. We show that Ctrl-G, where a 7B-parameter TULU2 model is combined with a 2B-parameter HMM, beats significantly larger LLMs like GPT4 on the task of generating text insertions/continuations with logical constraints. On commonly used constrained generation benchmarks like CommonGen, Ctrl-G beats other constrained generation approaches, as well as supervised training, by large margins. In addition to the dominant paradigm of prompt engineering, our work opens up new avenues for achieving tractable, reliable and fine-grained inference-time control of LLMs.

## Acknowledgments

This work was funded in part by the DARPA ANSR program under award FA8750-23-2-0004, the DARPA PTG Program under award HR00112220005, NSF grant #IIS-1943641, NSF CAREER Award #2339766, and AFOSR MURI via grant #FA9550-22-1-0380.

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

## A    Derivation of Eq. (4)

$$\boxed{p(S_n \in F \mid z_t, s_t)}$$

$$= \sum_{x_{t+1}, z_{t+1}} p(S_n \in F, x_{t+1}, z_{t+1} \mid z_t, s_t)$$

$$= \sum_{x_{t+1}, z_{t+1}} p(S_n \in F \mid x_{t+1}, z_{t+1}, z_t, s_t) \cdot p(x_{t+1}, z_{t+1} \mid z_t, s_t)$$

$$= \sum_{x_{t+1}, z_{t+1}} p(S_n \in F \mid S_{t+1} = \delta(s_t, x_{t+1}), z_{t+1}) \cdot p(x_{t+1} \mid z_{t+1}) \cdot p(z_{t+1} \mid z_t)$$

$$= \sum_{z_{t+1}} \sum_{s_{t+1}} \sum_{x_{t+1} \in \mathrm{edge}(s_t, s_{t+1})} p(S_n \in F \mid s_{t+1}, z_{t+1}) \cdot p(x_{t+1} \mid z_{t+1}) \cdot p(z_{t+1} \mid z_t)$$

$$= \sum_{z_{t+1}} p(z_{t+1} \mid z_t) \cdot \sum_{s_{t+1}} \boxed{p(S_n \in F \mid z_{t+1}, s_{t+1})} \cdot \sum_{x_{t+1} \in \mathrm{edge}(s_t, s_{t+1})} p(x_{t+1} \mid z_{t+1}).$$

## B    DFA operations

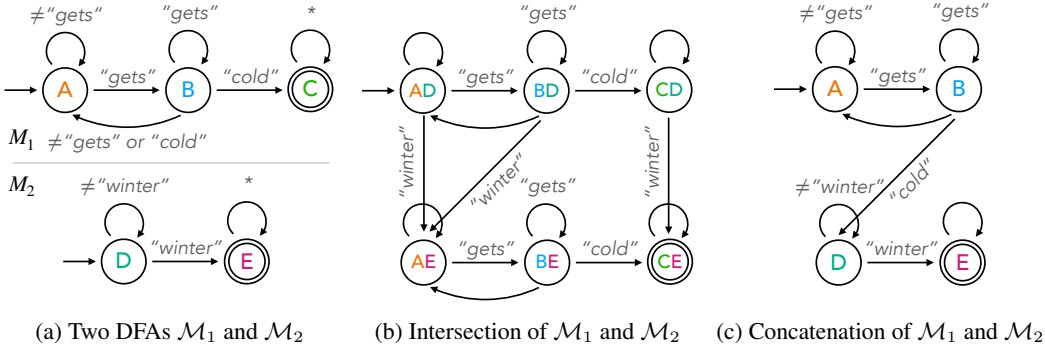

(a) Two DFAs $\mathcal{M}_1$ and $\mathcal{M}_2$     (b) Intersection of $\mathcal{M}_1$ and $\mathcal{M}_2$     (c) Concatenation of $\mathcal{M}_1$ and $\mathcal{M}_2$

Figure 6: An example showing the intersection (logical and) and concatenation of two DFAs.

**Proposition B.1.** *Let $\mathcal{M}_1$ be a DFA such that for each accept state $s$, $\delta(s, w)$ goes to a dead state for all $w \in \Sigma$. Then $\mathcal{M}_1$ can be concatenated with any other DFA $\mathcal{M}_2$ by merging the accept states of $\mathcal{M}_1$ with the initial state of $\mathcal{M}_2$;*

here a *dead state* denotes a DFA state that is (1) not an accept state and (2) only transitions to itself. Instead of formally defining what it means by "merging" the initial state of $\mathcal{M}_2$ with the accept states of $\mathcal{M}_1$, we refer readers to Figure 3c for such an example.

## C    Human evaluation

Table 5 presents the aggregated results for all questions from the Human Evaluation. Each question was answered by three workers, and we compute their inter-annotator agreement. Each worker evaluated the outputs generated by four different models for the same prefix (and suffix) within each batch. We converted these evaluations for each batch into rankings and then used the Kendall Coefficient of Concordance to assess the correlation between the rankings assigned by each worker. The average coefficient was 0.449, indicating a moderate level of agreement among the annotators.

You will be presented with short paragraphs ( A, B, C and D) extracted from some stories. Read each paragraph and rate them by answering the following questions.

**Paragraph A**

Dad, the ultimate judge and jury, is now being judged. Based on dad's values? Of course not! Why would anyone put their own needs and desires aside to stroke dad's ego? It is high time dad's It is high time dad's brainstorm a new approach to handle this growing trend of judgment. idea how to handle this growing trend of judgement.

**Questions**

(1) **Fluency**: Is Paragraph A well-formed and fluent?

○ 5: **Yes**, it is well-formed and fluent.
○ 4: Between 3 and 5
○ 3: **Somewhat**, there are a few interruptions, but understandable.
○ 2: Between 1 and 3
○ 1: **No**, the writing is completely broken.

(2) **Coherence**: Is Paragraph A consistent and logical?

○ 5: **Yes**, it is consistent and logical.
○ 4: Between 3 and 5
○ 3: **Somewhat**, there are a few rough transitions, but overall consistent.
○ 2: Between 1 and 3
○ 1: **No**, the story makes no sense.

(3) **Overall**: Based on your answers to (1) and (2), is Paragraph A overall well-written and logical?

○ 5: **Perfect**, it is well-written and logical.
○ 4: Between 3 and 5
○ 3: **Okay**, it has some issues, but is overall acceptable.
○ 2: Between 1 and 3
○ 1: **Poor**, the writing is broken and the story makes no sense.

**Paragraph B**

Dad, the ultimate judge and jury, is now being judged. Based on dad's values? Of course not! Why would anyone put their own needs and desires aside to stroke dad's ego? It is high time dad's values be judged? idea how to handle this growing trend of judgement.

**Questions**

(1) **Fluency**: Is Paragraph B well-formed and fluent?

○ 5: **Yes**, it is well-formed and fluent.
○ 4: Between 3 and 5
○ 3: **Somewhat**, there are a few interruptions, but understandable.
○ 2: Between 1 and 3
○ 1: **No**, the writing is completely broken.

(2) **Coherence**: Is Paragraph B consistent and logical?

○ 5: **Yes**, it is consistent and logical.
○ 4: Between 3 and 5
○ 3: **Somewhat**, there are a few rough transitions, but overall consistent.
○ 2: Between 1 and 3
○ 1: **No**, the story makes no sense.

(3) **Overall**: Based on your answers to (1) and (2), is Paragraph B overall well-written and logical?

○ 5: **Perfect**, it is well-written and logical.
○ 4: Between 3 and 5
○ 3: **Okay**, it has some issues, but is overall acceptable.
○ 2: Between 1 and 3
○ 1: **Poor**, the writing is broken and the story makes no sense.

Figure 7: Human evaluation interface on Amazon Mechanical Turk.

Table 5: Full human evaluation results.

| | Continuation | | | | Insertion | | | |
|---|---|---|---|---|---|---|---|---|
| | None | K | L | K&L | None | K | L | K&L |
| *Q1. Fluency* | | | | | | | | |
| TULU2 | 4.06 | 3.99 | 4.20 | 4.22 | 2.77 | 2.77 | 2.87 | 2.89 |
| GPT3.5 | 4.52 | 4.45 | **4.58** | **4.50** | 2.33 | 2.34 | 2.37 | 2.39 |
| GPT4 | **4.58** | **4.50** | 4.57 | 4.44 | 3.91 | 3.51 | 3.66 | 3.23 |
| Ctrl-G | 4.31 | 4.23 | 4.42 | 4.22 | **4.00** | **3.80** | **4.02** | **3.90** |
| *Q2. Coherency* | | | | | | | | |
| TULU2 | 3.92 | 3.89 | 3.95 | 3.96 | 2.82 | 2.84 | 2.96 | 2.98 |
| GPT3.5 | 4.54 | 4.43 | **4.55** | **4.46** | 2.60 | 2.48 | 2.57 | 2.62 |
| GPT4 | **4.59** | **4.54** | 4.53 | 4.37 | **3.90** | 3.49 | 3.75 | 3.32 |
| Ctrl-G | 4.23 | 4.04 | 4.38 | 4.05 | 3.88 | **3.68** | **3.78** | **3.67** |
| *Q3. Overall Quality* | | | | | | | | |
| TULU2 | 3.80 | 3.77 | 3.87 | 3.88 | 2.68 | 2.64 | 2.78 | 2.74 |
| GPT3.5 | 4.40 | 4.32 | **4.44** | **4.36** | 2.27 | 2.22 | 2.27 | 2.31 |
| GPT4 | **4.48** | **4.44** | **4.44** | 4.26 | **3.79** | 3.33 | 3.53 | 3.10 |
| Ctrl-G | 4.13 | 3.98 | 4.27 | 3.96 | **3.77** | **3.56** | **3.73** | **3.59** |

Table 6: Prompt templates for querying the GPT3.5 and GPT4 models on the task of text editing.

**Continuation:**

Below is the opening of a story. Continue the narrative by writing the next few sentences that includes the specified keywords. Your continuation should naturally follow the themes, tone, and setting established in the opening. Aim to write a compelling and coherent continuation that could lead the story forward. Your answer must consist of at least (WordRangeStart) words and no more than (WordRangeEnd) words. Please make sure to incorporate the given keywords in to your answer. Keywords: (Keyword).

Story: (Prefix)

**Insertion:**

Fill in the text at the [INSERT] in the following story with an appropriate sentence that includes the specified keywords. Feel free to use your knowledge, guesses, or interpretations to craft your answer, but ensure it is relevant to the context provided by the prefix and suffix. Your answer must consist of at least (WordRangeStart) words and no more than (WordRangeEnd) words. Please make sure to incorporate the given keywords in to your answer. Keywords: (Keyword).

Story: (Prefix)[INSERT](Suffix)

