# OpenReview forum: "Adaptable Logical Control for Large Language Models"
_NeurIPS.cc/2024/Conference — NeurIPS 2024 poster_

### Official Review · Reviewer_eXgv · 2024-07-08

**Soundness:** 3
**Presentation:** 2
**Contribution:** 3
**Rating:** 6
**Confidence:** 3

**Summary:**

This paper proposes an approach called Ctrl-G to control LLM generation, specifically, constraining LLM’s output to deterministically follow certain logical constraints, such as maintaining a certain keyword in the generated text.
The approach has two main parts, the first is a Hidden Markov Model (HMM) that serves as a “prediction” model to guide the generation. The data used to train the HMM is sampled from the LLM. Essentially, the HMM is like a distillation model of the LLM that tries to capture the next-token generation “search space” of the LLM on a specific task. The second part is a series of deterministic finite automata (DFAs), each designed to represent a certain logical constraint $\alpha$. The DFAs serves as a “checker” to determine the acceptance or rejection of an output from LLM. Combining the HMM and the DFAs, the proposed approach computes the conditional probability $P_hmm(\alpha | x_t, x_<t)$, i.e., how likely $x_t$ is leading to $\alpha$ being satisfied, to sample the next token.

**Strengths:**

The idea of using a HMM model to approximate a white-box model for exploring the LLM’s token generation space is novel. Despite that the idea of using HMM was published in a previous work [Zhang et al. Tractable control for autoregressive language generation], this paper proposes to improve by using DFAs for modeling various other task-specific logical constraints.

**Weaknesses:**

The evaluation requires more details and/or more experimental results to justify the contribution of the proposed approach. The limitation of the proposed approach is not clearly laid out and requries more discussions. Please see questions and limitations below.

**Questions:**

In section 4, the proposed approach is compared with FUDGE [Yang and Klein, 2021], NeuroLogic A*esque decoding [Lu et al., 2022], and ILM model [Donahue et al. 2020]. The background and motivation of comparing with these baseline methods is not clear. Why choose these models for comparison?

In section 5, the proposed approach is compared with plain GPT-3.5-Turbo and GPT4-Turbo. The comparison should also be made using GPT models and the proposed DFAs method as an output checking method. In fact, there are other simpler constrained parsing techniques, for example, the method used in [Constrained Language Models Yield Few-Shot Semantic Parsers](https://aclanthology.org/2021.emnlp-main.608) (Shin et al., EMNLP 2021).

In section 5, the HMM is distilled from TULU-2-7B model. Is there a specific reason for choosing this model? A 7B model seems relatively small. The llama-2 series and Google Gemma has open-sourced models with larger (> 20B) parameter sizes.

Since distillation of a HMM is the key in this approach, the HMM’s training quality should be a concern. The paper mentioned that the HMM was trained on millions of samples generated from a baseline LLM (not fine-tuned on the task-specific data), but if those samples are of bad-quality, how can the quality of the obtained HMM be assured?

The paper claimed a 100% constraint satisfaction achieved with the proposed approach. Following the previous three questions, it is unclear if this 100% is achieved with the HMM model or the constrained parsing with DFAs. The proposed approach would be more valuable if the author can demonstrate the contribution is mainly from the HMM.

**Limitations:**

As mentioned in the questions, the experiment only shows the HMM being trained on a 7B model. Is it possible to generalize the approach to larger models, eventually to be used with models with trillions of parameter like GPT? For example, is the sampling amount a barrier for distillation from larger models? If so, what is the size limit here? More discussion should be made on the benefits of the proposed approach.

---

> ### Author Rebuttal · Authors · 2024-08-06
>
> Thank you for your feedback and questions.
>
> We would like to clarify that Ctrl-G does not use DFAs to post-check the generated text; instead, the generated text is always guaranteed to satisfy the constraint. Specifically, Ctrl-G achieves constrained generation by generating from
> 	$p_{ctrlg}(x_{t} | x_{<t}, \alpha) \propto p_{lm}(x_{t} | x_{<t}) p_{hmm} (\alpha | x_{\leq t})$,
> where the LM $p_{lm}(x_{t} | x_{<t})$ is responsible for generating high-quality text while the HMM $p_{hmm} (\alpha | x_{\leq t})$ is responsible for guiding the LM to satisfy the constraint $\alpha$. Here, we assume that $\alpha$ can be represented as a DFA so this marginal probability $p_{hmm} (\alpha | x_{\leq t})$ can be computed by the algorithm proposed in Sec. 3.2.
>
> ### **Choice of baselines:**
>
> For the CommonGen benchmark, we chose FUDGE as a representative for the classifier-guided constrained generation methods; we chose NeuroLogic A\*esque as a representative for the search-based decoding methods. We have further included NADO and GeLaTo as the baselines and we show that Ctrl-G beats them by large margins. **Please refer to the global response for more details.** We have also released the outputs of Ctrl-G via an anonymous link shared to the AC.
>
> For the task of text infilling, ILM is the only available baseline that can generate infillings for multiple masks of arbitrary length; all baselines for CommonGen cannot be immediately adapted to this particular task. Nevertheless, given that Ctrl-G is an unsupervised approach, ILM, which is finetuned with full supervision on the task of text infilling, is a very strong baseline.
>
>
> ### **Quality of the distilled HMMs:**
>
> By training an HMM on examples sampled from the base LM, our goal is to effectively minimize the KL-divergence between $p_{hmm}$ and $p_{lm}$ and the actual quality of the samples should not matter. In the ideal case, we would have $p_{hmm} = p_{lm}$ and using Ctrl-G would be equivalent to sampling from $p_{lm} (x_{1:n} | \alpha)$. Note that Ctrl-G controls the generation from the base LM by approximating $p_{lm} (x_{t} | \alpha, x_{< t})$ but is not expected to extrapolate beyond the distribution of the base LM. Luckily, given the recent advancement of LLMs, the quality of the base LM should not be a bottleneck.
>
> ### **Generalize to larger LLMs:**
>
> Though there are larger open-source LLMs, we chose a 7B-parameter model due to the limit of computation. However, this is not a weakness of this work:
>
> - Note that prior controllable generation approaches like NADO, FUDGE, GeLaTo and NeuroLogic A*esque were only evaluated on LMs with < 0.7B parameters; by demonstrating the effectiveness of Ctrl-G on a 7B-parameter LLM, we have provided strong evidence for scaling it up to even larger models with potentially hundreds of Billions of parameters.
>
> - It would be more computationaly expensive to distill an HMM from larger LLMs but it would not be a bottleneck. (1) The distillation process is independent from the constraints: once the HMM is trained, it can be used to enforce whatever constraints that can be represented as DFAS. (2) According to our experience with GPT2-large and TULU2-7B, it turns out that we don't really need to sample a lot more examples from larger LLMs: 128 million tokens for GPT2-large and 320 million tokens for TULU2-7B are good enough.
>
> - We show that Ctrl-G allows an LLM with *only 7B parameters* (coupled with a 2B-parameter HMM) to beat GPT3.5 and GPT4. This constitute an even stronger argument demonstrating the effectiveness of our approach, compared to showing that Ctrl-G allows a 70B-parameter LLM to beat the GPT models.
>
> We will add a more detailed discussion to the paper.

---

> > ### Comment · Reviewer_eXgv · 2024-08-12
> >
> > Thank you for the reply. The rebuttal addresses some of the questions.
> >
> > I understand that Ctrl-G does not use DFAs to explicitly post-check the generated text. However, it remains unclear how Ctrl-G compares to simpler constrained decoding methods, such as those described in "Constrained Language Models Yield Few-Shot Semantic Parsers" (Shin et al., EMNLP 2021). This is especially the case when these simpler methods are combined with more powerful LLMs like GPT-4.
> >
> > As noted in the review, the evaluation only compares Ctrl-G to plain GPT-4 models. Given this, the contribution of the proposed approach could be questioned. The added value of achieving comparable (or even inferior) performance to GPT-4 with a simpler decoding method, while requiring significantly more training effort to develop a HMM, is not entirely clear.
> >
> > I agree with reviewer N2WB that the section describing DFAs is too lengthy and could be condensed. As noted in my review, the discussion on distilling an HMM seems to be the major contribution, while the DFAs provide only incremental improvements. And training details such as using the KL divergence should be briefly mentioned to give more context about the unsupervised learning. However, this also may make the paper somewhat incremental compared to the GeLaTo work.
> >
> > It would be beneficial if the authors could clarify the differences between GeLaTo and Ctrl-G in the methodology section. Specifically, highlight the contributions of introducing DFAs and demonstrate clearly in the evaluation results.
> >
> > The main concern regarding the contributions of the work has not been fully addressed, so I will maintain my current rating.

---

> ### Author Response · Authors · 2024-08-14
> **Further Clarification**
>
> Thank you for following up with our discussion. Here are some further clarifications to your questions.
>
> ----
>
> ### **Ctrl-G vs. GPT4.**
> We would like to first clarify that Ctrl-G (with TULU2-7B and HMM-2B) actually **beats GPT4 by large margins**. Please refer to the pdf shared in our global response for details: the *Overall* section of Table 3 measures the percentange of examples where the models' outputs (1) satsifies the given constraints (keyphrase inclusion, word count range) **and** (2) attains an average quality score higher than 3 (out of 5) in human evaluation. Here **Ctrl-G beats GPT4 by 30% - 70% in Overall Satisfaction rate in all settings.**
>
> ----
>
>
> ### **Why do we need an HMM?**
> Thank you for mentioning [1]. To the best of our understanding, [1] mainly leverages SCFGs to prune away the next-token that would violate the constraints. From this perspective, [1] is actually similar to *Outlines* [2] and *guidance* [3], which also use CFGs/DFAs to prune away the next-tokens.
>
> As discussed in Sec. 3.3, **[1, 2, 3] is subsumed by Ctrl-G in the sense that [1, 2, 3] only decide whether $p_{hmm}(\alpha | x_{\leq t})$ is 0 or not.** [1, 2, 3] would not work for many applications because they do not have any probabilistic information. We added the following example to Sec. 3.3 to illustrate the distinction: consider the task of generating a sentence that ends with the phrase " in the park", where we compare Ctrl-G with guidance, both applied to the same model.
>
> ```
> guidance: silhouette of suspected ... an heavily secured.in the park
> Ctrl-G: A man and a woman are walking in the park
> ```
>
> Even though both generations end with " in the park", it is clear that the output from guidance is not desirable as it unnaturally appends the phrase to some irrelevant text. The reason is that guidance, by performing *pure logical reasoning* over the DFA, only discard the next tokens that would make the constraint unsatisfiable, while the probabilities of the other next tokens remain unchanged; in contrast, Ctrl-G performs *probabilistic reasoning* by estimating $p_{lm}(\alpha | x_{\leq t})$; i.e., we estimate how likely each next token would eventually lead to $\alpha$ being satisfied.
>
> Hopefully this would answer your question why an HMM is needed. We have added [1] as a reference to Sec. 3.3. Thank you for your suggestion.
>
> ----
>
> ### **HMM training details:**
> The goal of the HMM training is to minimize $D_{KL}(p_{lm} || p_{hmm}) = E_{x_{\leq n} \sim p_{lm}} \log p_{lm}(x_{\leq n}) - E_{x_{\leq n} \sim p_{lm}} \log p_{hmm}(x_{\leq n})$, which is equivlant to maximizing the second term (log-likelihood of the data sampled from the LM) because the first term (entropy of the LM) is a constant. We will add a more detaile discussion to the methodology section. Thank you for your suggestion.
>
> ----
>
> ### **Ctrl-G significantly generalizes GeLaTo**:
>
> GeLaTo proposes the idea of using a distilled HMM to approximate  $p_{lm}(\alpha | x_{\leq t})$ via $p_{hmm}(\alpha | x_{\leq t})$, which is indeed significant, but **GeLaTo is not applicable to many down-stream applications**: it only derived the algorithm for computing the marginal probability $p_{hmm}(\alpha | x_{\leq t})$ for $\alpha$ being the keyword constraints. i.e., **GeLaTo could not handle the keyword exclusion, text infilling or word count control** because the algorithm for computing such marginal probability in general is not known in existing literature.
>
> Compared to Ctrl-G, the main technical contribution of Ctrl-G is deriving a polynomial-time and GPU-parallelizable algorithm for computing $p_{hmm}(\alpha | x_{\leq t})$, as long as $\alpha$ can be represented as a DFA (see Sec. 3.2). This contribution is not trivial. We will add more technical details (e.g., deriviation of Eq. 4) and show how the algorithm is tensorized in the revised paper. Besides, we have also cut down the DFA content by over a half to highlight our technical contribution here. Thank you for your suggestion.
>
> In summary, **GeLaTo only supports the down-stream application of keyword-constrained generation, and Ctrl-G significantly generalizes GeLaTo by allowing this approach to be applicable to arbitrary constraints that can be represented as DFAs.**
>
> We will add a more comprehensive comparison between GeLaTo and Ctrl-G to the methodology section. Thank you for your suggestion. Please let us know if you have further questions or concerns.
>
> ---
>
> [1] Shin, Richard, et al. "Constrained Language Models Yield Few-Shot Semantic Parsers." Proceedings of the 2021 Conference on Empirical Methods in Natural Language Processing. 2021.
>
> [2] Brandon T Willard and Rémi Louf. Efficient guided generation for large language models. arXiv e-prints, pages arXiv–2307, 2023.
>
> [3] Scott Lundberg, Marco Ribeiro, Richard Edgar, and Harsha-Nori. Guidance: a guidance language for controlling large language models., 2024.

---

### Official Review · Reviewer_Uqfm · 2024-07-11

**Soundness:** 3
**Presentation:** 2
**Contribution:** 3
**Rating:** 6
**Confidence:** 3

**Summary:**

This paper proposes Ctrl-G to enable flexible lexically constrained generation with high accuracy and efficiency. Ctrl-G first distills an HMM from an unconditioned language model and then formulates the lexical/logical constraints with deterministic finite automata (DFAs). The inference algorithm takes both the HMM and the DFA as input and computes the desired conditional probability for guiding 48 LLM generation towards satisfying the given constraints. Experiments in keyphrase generation, text infilling, interactive text editing, and mathematical reasoning indicate that Ctrl-G achieves competitive performances for various sizes of LMs.

**Strengths:**

1. The proposed method is flexible and manageable for constrained generation. It achieves competitive performance on various tasks, and it can scale up to control larger models with high efficiency.
2. The proposed method is theoretically sound and is clearly presented. The illustrations help in understanding the DFA formulation.

**Weaknesses:**

1. The experimental comparison for small model tasks is relatively weak. The paper discussed several related works in the introduction and method part (e.g., GeLaTo and NADO), but the works are not included in the experiments. The evaluation metrics BLEU and ROUGE-L measure text overlaps and cannot comprehensively represent the generation quality. Qualitative results are not provided.
2. The experiment part of the paper is a little bit hard to follow. Section 5 and 6 represent key improvements, but the experimental results are displayed in the appendix (Table 4 and 5). Some sentences/sections are not finished.

**Questions:**

See weakness.

**Limitations:**

The authors discuss limitations in the experiment sections but do not discuss potential negative societal impact.

---

> ### Author Rebuttal · Authors · 2024-08-06
>
> Thank you for your feedback and questions.
>
> - For the CommonGen benchmark, we did not include NADO because they only published results on the dev set and their released code is not reproducible due to lack of documentation; In the revised paper, we have added the results from NADO and GeLaTo on both dev & test sets, along with more evaluation metrics, for a direct comparison. **Please see the global response for the updated results.**
>
> - Regarding qualitative examples, for a quick reference, here are the outputs of Ctrl-G on the first 10 examples of the dev set of CommonGen; each line is a list of concepts followed by the output:
> ```
> field stand look: a woman stands in a field looking at flowers
> kid room dance: The kids are dancing in the living room.
> pet couch cat: A pet cat sleeping on the couch.
> climb side building: A man climbs up the side of a building.
> talk climb wall: A man climbs a wall while talking on a cell phone
> snow car drive: A car drives through the snow.
> phone talk wear: A man wearing sunglasses talks on a cell phone.
> rink hockey team: hockey player on rink during team practice
> surfer surf ocean: surfers surf in the ocean off the coast
> flight stair jump: A dog jumps down a flight of stairs.
> ```
> We have further released the outputs of Ctrl-G for all three tasks, i.e., CommonGen, Text Infilling and Interactive Text Editing. **Following the instructions for authors, we uploaded them to an anonymous link and shared the link with the AC.**
>
> - For the text infilling task, ILM, as a model trained with full supervision, is already a very strong baseline for Ctrl-G, which is not trained with any supervision on infilling. Furthermore, all baselines for CommonGen cannot be adapted to this task. In addition, to the best of our knowledge, the diffusion-based LMs cannot be used to fill in multiple blanks of unknown length thus cannot be adapted to this task either.
> - In the revised paper, we cut down the contents about DFAs by more than a half and moved all main results, pseudocode for the algorithm, and the runtime analysis from the appendix back to the main paper. We have also cleaned up the unfinished sentences and greatly improved the writing. We would be more than happy to share the revised version upon AC’s approval.
> - Potential social impact: as we discussed in Sec. 6, Ctrl-G could potentially be used to detoxify language models by preventing inappropriate phrases from appearing. At the same time, it is also possible that people can use Ctrl-G to make LM generate toxic contents by enforcing the occurrence of inappropriate contents. The ability to control the generation from LMs is indeed a double-edged sword and we will add a more detailed discussion on the potential social impact of our work.

---

> > ### Comment · Reviewer_Uqfm · 2024-08-11
> >
> > I appreciate the authors' detailed response. The rebuttal addresses my concerns and I will keep my rating.

---

### Official Review · Reviewer_GDqt · 2024-07-13

**Soundness:** 4
**Presentation:** 4
**Contribution:** 4
**Rating:** 7
**Confidence:** 3

**Summary:**

It is a challenge to control LLMs' generation to adhere to logical constraints. Ctrl-G integrates LLMs with a Hidden Markov Model (HMM) and deterministic finite automata (DFAs) for flexible and tractable control, such as keyword and length constraints. Ctrl-G combined with TULU-2-7B model outperforms GPT-3.5 and GPT-4 in human evaluations for interactive text editing, achieving a 30% higher satisfaction rate and 100% constraint satisfaction. Additionally, Ctrl-G improves performance on the Grade School Math (GSM) dataset by guiding the reasoning process with logical constraints.

**Strengths:**

1. The paper presented an interesting method on logically-controlled text generation.
2. Comprehensive experimental results and analysis are provided. The results are strong and the findings are backed by well-designed human evaluation. Runtime analysis on efficiency of the method is provided.
3. The paper is well-written and presented.

**Weaknesses:**

There are no major weaknesses.

**Questions:**

1. Can the framework potentially handle highly creative tasks that might require more fluid and less structured outputs?
2. Can Ctrl-G potentially generalize to domains outside of those tested, such as scientific text generation or code synthesis? No need to provide results, but insights on this would be helpful.

**Limitations:**

Yes, the authors adequately addressed the limitations

---

> ### Author Rebuttal · Authors · 2024-08-06
>
> Thank you for your feedback and questions.
>
> - We believe that better control in general will substantially benefit creative tasks. For example, according to the experience of our collaborators, when LLMs are asked to generate generic lyrics, they often tend to generate lyrics for love songs even when they are instructed not to do so. We can potentially avoid such a phenomenon by using Ctrl-G to prevent the phrases commonly used by love songs from appearing.
> - The effectiveness of Ctrl-G mainly relies on how well the assumption $p_{hmm}(\alpha | x_{\leq t}) \approx p_{lm}(\alpha | x_{\leq t})$ holds. Distributions over scientific writing or code could be potentially harder to approximate but there should not be a fundamental difference compared to generic writing: for the more challenging domains we can always further scale up the HMMs.

---

### Official Review · Reviewer_N2WB · 2024-07-18

**Soundness:** 2
**Presentation:** 1
**Contribution:** 3
**Rating:** 5
**Confidence:** 3

**Summary:**

The paper tackles the problem of generating sequences from llms while following a constraint \alpha, providing a solution for the case where \alpha can be represented as a regular language (e.g., a constraint on containing certain substrings).  The method allows application of any regular-language constraint, and is somewhat similar to/inspired by a previously proposed method, Gelato, in which the LLM is distilled into an HMM.

The method is evaluated with domain-finetuned TULU-7B language models, and compared to some other constrained sequence generation methods, FUDGE and Neurologic A* esque decoding: these results are favourable. It is also compared to baselines such as directly using very large language models (e.g. GPT4), with the constraint given simply as part of the input. These results are unfortunately missing (the appendix is missing), but apparently GPT4 struggles to follow constraints (whereas ctrl-G always follows the constraints, by design). It is not clear to me whether the method was compared to Gelato.

A small investigation is done to see whether constrained decoding can help when solving math problems, specifically by forcing the solution to contain all numbers from the question, and the method is deployed on the GSM benchmark. The performance is improved relative to not using the method, but not compared to other methods.

**Strengths:**

1. A method for constrained generation that always follows the constraints, by construction
2. The method possibly allows richer constraints than a previous one? Unfortunately I am not entirely sure about this, it is not clearly enough stated in the work
3. Comparison with some previous methods

**Weaknesses:**

1. Weak evaluation (see questions/comments), in particular no comparison to gelato (?), no inference time comparison ( I get the impression that inference with this method is very slow), key results put in appendix, GSM experiment more a demonstration of usefulness of constrained decoding than of ctrl-G in particular (no comparison to other other constrained decoders)
2. Some hard to follow details and overview. Missing in particular: 1) a complete description of how inference is done in this framework, and 2) more detailed description of differences and similarities between this method and Gelato (both use an HMM distilled from the LLM, but Gelato then uses only that while this method still uses the LLM as well - why does this method have both an LLM and its distilled HMM - I will appreciate more intuition on what is happening here). Also, it is not clear how/whether this method differs from gelato with respect to inference complexity (time) or potential constraint space (i.e. type of constraints that can be encoded)
3. Confusing amount of attention given to problem of expressing "regular" constraints (e.g., contains/doesnt contain substring, has maximum length 3) as DFAs, I would expect this is trivial, but somehow the authors treat it as not, referencing multiple algorithms and dedicating several paragraphs and figures (e.g. 3: example of a DFA, 4: operations on DFAs) to it. 2 figures and over 40 lines (127-135, 176-196, 222-235) dedicated to question of creating DFAs for rather simple constraints, all would be better spent on describing main alg and bringing in main results (which are currently in missing appendix).

**Questions:**

questions/comments:

0. several references made to materials in appendix, but no appendix! even if it were there, main materials should not be in appendix - pseudo code and greater description of ctrl g would be much more appreciated in main, as well as several results referred to from main text!
1. figure 2 never referenced in main text
2. lines 38-39 unfinished/mangled sentence
3. line 39-40: possibly unclear sentence? it reads like gelato provides multiple algorithms in general, but for the case of keyword constraints it only provides one, in which case it is unclear what this alg is, what the situation is for other types of constraints, and whether ctrl-G covers the same or a larger set of constraints as can be represented with gelato. generally, the comparison between ctrl-g and gelato is not clear enough on the similarities and differences and in fact on what gelato is exactly. end of this sentence (in line 41) also unclear.
4. lines 87-91: it seems (also from lines 39-41) that the paper faults Gelato for only providing one algorithm for handling certain constraints. but doesn't this paper also have only one algorithm? this criticism doesnt make sense
5. lines 87-91: gelato is critiqued for possibly not being efficiently computable, but my impression is that this is true for ctrl-g too (with complexity nmh^2, where h is the distilled HMM size and reaches 32k even in this paper). can you compare the two explicitly?
6. lines 87-91: the claim that it is unclear whether gelato would scale - why is that? is it because of the distillation to an hmm, and an assumption that this will become a worse approximation as the original model grows larger? if so, does that drawback not hold also for this method?
7. lines 99-111: the description says a DFA is used to encode the constraint, but the DFA does not seem to appear in the equations, and it is not clear how it factors in
8. line 121 typo: "fo"
9. line 125 consider making it clearer that q0 here is "the" q0
10. 127-135 this feels overly mystical for the problem of designing a dfa to express simple constraints. Unless I am missing something (in which case I would appreciate explanation), I think the whole discussion of designing DFAs for the constraints does not need more details than: "this method allows for any constraint that can be expressed with a DFA", with maybe a reference to textbooks on dfas if someone is not experienced. similarly figs 3 and 4 are not necessary.
11. line 138 is this DFA only partial? if not, surely m=k * |\Sigma| - why not just write that?
12. 146 what subscript?
13. 146-151 here i could do with being walked through what is happening in simpler terms/more detail/justifications/something - this bit is harder for me
14. thrm 3.2 - give some justificution. did you compute this? (if so, prove). is it taken from a paper? if so, refer.
15. line 166: "we set": i get what youre saying but rephrase because you cant set this distribution as you please. maybe you mean used 0/1 or only evaluated whether or not phmm(a|..)>0
16. section 4 again bewildering amount of space dedicated to design of dfas for ultimately very simple constraints. line 170, and then lines 176-196 (also, 186-189 seem to be a definition not a theorem), all for this simple task. Also, if must refer to various algorithms (e.g. Aho-Corasick, line 182), then need to also provide relevant sources (ie cite).
17. lines 202-204 im a bit confused: if each epoch has its own samples, what separates epochs? i dont understand. i see similar phrasing used in gelato but i dont follow this. are you changing anything between epochs? if not, this sounds like 1 epoch over 4m samples? if yes, explain what is being changed
18. line 226 "inflection of keywords" - huh?
19. 222-235 (and 235-241 kind of): more space spent on needless discussion of dfa operations. i respect that maybe sometimes these operations are computationally expensive but it seems all constraints considered in this paper do not bump against this anyway. it would make more sense to mention this only if it somehow limited the experiments or practical applications of the method.
20. line 231: "dead states" not been defined
21. line 259: seems like a fairer comparison against ILM would have it trained with the right percentage of masked out tokens each time, or even a varying percentage if want to see how it performs when only allowed one model
22. line 300 "word range" -  maybe you mean sentence length range? word count range?
23. line 331: saying ctrl-G is a 7B param model: you are ignoring the parameter count of the distilled HMM, which seems to me quite large (32k states!). What is the vocabulary size of Tulu2? If I assume 50k (i.e. similar to GPT2), that's 32 * 50M = 1.6B parameters for the emission matrix, and another 32 * 32 M = ~1B parameters for the transition matrix.
24. line 339-341 hard to understand what being compared to what, rephrase
25. 350-352: everything following "we can" - it is possible, but this is speculation, and not backed by the results, so be more careful in presentation.

**Limitations:**

weak eval - missing comparisons and results

---

> ### Author Rebuttal · Authors · 2024-08-06
>
> Thank you for your detailed feedback. Following some prior years’ tradition, our appendix was mistakenly uploaded as the supplementary material and we apologize for the confusion. We have greatly improved the presentation of the paper and would love to share the revision upon AC’s approval.
> - We cut down the DFA content by more than a half and we will further move Fig. 3 & 4 to the appendix as you suggested.
> - The pseudocode, the main experiment results and the runtime analysis is moved from the appendix to the main text. **Please see the global response for the main results.**
>
> ### **Efficiency of Ctrl-G:**
>
> Despite the time complexity result $O(nmh^2)$ from Theorem 3.2., Ctrl-G is actually quite fast due to its GPU-based implementation, where the $mh^2$ part is fully tensorized. For example, on the task of interactive text editing, Ctrl-G, when applied to the TULU2-7B model, is able to generate infillings under multiple constraints within a few seconds, enabling a user interface for real-time interaction. On the CommonGen benchmark, in the unsupervised setting, Ctrl-G not only generates outputs of higher quality but also runs 30 - 100 times faster than NeuroLogic A\*esque and 6 - 16 times faster than GeLaTo. **Please refer to the global response for details.**
>
> ### **Comparison between GeLaTo and Ctrl-G:**
>
> GeLaTo is really a special case of Ctrl-G. GeLaTo is much more limited in the sense that it only supports the keyword constraint, that is, no support for text infilling, word count control or arbitrary DFAs.
>
> Both GeLaTo and Ctrl-G follow the same formulation for controllable autoregressive generation: they sample from $p_{ctrlg}(x_{t} | x_{<t}, \alpha) \propto p_{lm}(x_{t} | x_{<t}) p_{hmm} (\alpha | x_{\leq t})$, where $p_{hmm} (\alpha | x_{\leq t})$ approximates $p_{lm} (\alpha | x_{\leq t})$; specifically, *both GeLaTo and Ctrl-G use the base LM and the distilled HMM*. The intuition is that $p_{lm}(x_{t} | x_{<t})$ is responsible generating fluent text while $p_{hmm}(\alpha | x_{\leq t})$ is responsible for guiding the LM to satisfy the constraint $\alpha$.
>
> We revised line 87 - 93 as:
>
> > - To control LM generation with an HMM, we need to compute $p_{hmm}(\alpha | x_{\leq t})$. GeLaTo only shows how to compute it for $\alpha$ being the keyword constraint. A polynomial-time algorithm for computing this probability for logical constraints in general is not known.
> > - GeLaTo assumes that $p_{hmm}(\alpha | x_{\leq t}) \approx p_{lm}(\alpha | x_{\leq t})$ and have verified its effectiveness on GPT2-large. It remains to be explored whether this assumption would hold for the more recent LLMs (e.g. Llama2) with over 10 times more parameters.
>
> > (1) In Ctrl-G, we propose a polynomial-time algorithm that computes $p_{hmm}(\alpha | x_{\leq t})$ **for any $\alpha$ that can be represented as a DFA.** (2) In addition to LMs on the scale of GPT2-large, we further verify the effectiveness of Ctrl-G in controlling LLMs with 7B parameters, demonstrating its scalability to even larger models.
>
> ### **Derivation of Eq. (4) and Theorem 3.2:**
>
> The derivation of Eq. (3) and (4) relies on the Markov property of HMMs and DFAs, and the fact that s_{t} is determined by x_{\leq  t}. We derived Theorem 3.2 and here is a sketch analysis:
>
> > The computation cost of Ctrl-G is dominated by the computation of $p(S_{n} \in F | z_{t}, s_{t})$ for all $t$, $z_t$ and $s_t$ following Eq. (4). Since $\sum_{x_{t+1} \in \text{edge}(s_{t}, s_{t+1})} p(x_{t+1} | z_{t+1})$ does not depend on t, we can precompute and cache it, resulting a cost of $O(mh|\Sigma|)$.
>
> > Then, note that for $s_{t}$, we only need to consider the $s_{t+1}$ where $\text{edge}(s_t, s_{t+1}) \neq \emptyset$. Hence, fixing $t$ and $z_t$, when we compute $p(S_{n} \in F | z_{t}, s_{t})$ for all $1 \leq s_t \leq k$, we only need to (1) enumerate through $1 \leq z_{t+1} \leq h$ and (2) for each $z_{t+1}$, we only need to visit each edge exactly once; there are $m$ edges in total, so it follows that cost is $O(n\cdot h\cdot h \cdot m) = O(nmh^2)$.
>
> > The total time complexity is $O(nmh^2 + mh|\Sigma|)$, which simplifies to $O(nmh^2)$ given that $|\Sigma| < nh$ in practice.
>
> ### **Comparison with ILM:**
>
> For the task of text infilling, the base LM used by Ctrl-G is not finetuned with any supervision on this task; it is only finetuned to adapt to the style of ROC stories. In contrast, the ILM baseline is explicitly trained with full supervision on text infilling. Hence, it is remarkable for Ctrl-G to match the performance of ILM on the training distribution of ILM (13% ratio). The evaluation on other masking ratios is meant to show that Ctrl-G is able to generalize well to different distributions.
>
> ### **Answers to Other Questions:**
>
> Line 99 - 111. We assume that $\alpha$ can be represented as a DFA so that $p_{hmm} (\alpha | x_{\leq t})$ can be computed by the algorithm proposed in Sec. 3.2.
>
> Line 138. We merge all transitions from one state to the other into one edge and the edges of a DFA denote the pair of states $(s_1, s_2)$ connected by transitions. Definition added to Sec. 3.1.
>
> Line 146. Changed to “we omit the subscript “hmm” for simplicity.”
>
> Line 166. Changed to “the other approaches are subsumed by Ctrl-G in the sense that they only evaluate whether $p_{hmm}(\alpha | x_{\leq t})$ is 0 or not”
>
> Line 202-204. “one epoch” is actually “one step of EM parameter update;” we will rephrase to avoid confusion.
>
> Line 226. “Inflections of keywords” refers to CommonGen where “swims” “swimming” are inflections of “swim”; for the task of text infilling we don’t need to handle such cases. We will remove this reference.
>
> Line 231. Dead states now defined in Sec. 3.1.
>
> Line 300. Changed to “word count range”
>
> Line 331. The vocab size of TULU2-7B is 32K; changed to “we show that Ctrl-G, where a 7B-parameter model is coupled with a 2B-parameter HMM, …”
>
> Line 350 - 352. “we can” changed to “In future work, it is possible to”

---

> > ### Comment · Reviewer_N2WB · 2024-08-13
> > **thank you**
> >
> > thank you for your comments and for sharing some missing results. i am surprised by how much faster ctrl-g is than gelato, this was not at all clear to me from the original manuscript! i raise my score. please do make all the changes needed for clarity as mentioned in my review. regarding latency, please clarify why ctrl-g can use the gpu but gelato cannot (if indeed it cannot. if it can, and just hasn't been implemented to take advantage of the gpu, please share this fact).

---

> > > ### Comment · Reviewer_N2WB · 2024-08-13
> > > **note - presented manuscript does need improvement!**
> > >
> > > please note that this accept is under the assumption that indeed the clarifications and presentation improvements have been made, as i cannot see the updated version here!

---

> > > > ### Author Response · Authors · 2024-08-14
> > > > **follow up**
> > > >
> > > > Thank you for following up with our discussion and increasing your rating.
> > > >
> > > > To the best of our knowledge, GeLaTo's implementation does make use of GPUs. However, the GeLaTo algorithm seems to be somewhat sequential thus is not fully parallelizable by GPUs.
> > > >
> > > > Regarding the paper revision, please be assured that we have already greatly improved the presentation of the paper and have further incorporated your suggestions. Thank you.
> > > >
> > > > We are also very eager to share our updated manuscript. However, according to the instruction for authors on the official website of NeurIPS 2024, we are not allowed to share it.
> > > >
> > > > ```
> > > > Authors may not submit revisions of their paper or supplemental material, but may post their responses as a discussion in OpenReview. This is to reduce the burden on authors to have to revise their paper in a rush during the short rebuttal period.
> > > > ```
> > > >
> > > > We have emailed the PC multiple times about possibilities of making an exception for us but we have not heard back yet. We'll keep reaching out to the AC and PC and let you know once we are allowed to share the updated paper.

---

> > > > > ### Comment · Reviewer_N2WB · 2024-08-14
> > > > > **Gelato parallelisation**
> > > > >
> > > > > Thanks. In your revised version, please make the effort to more thoroughly understand and explain why gelato inference is slower than ctrl-g, beyond just “seems sequential”!

---

### Author Rebuttal · Authors · 2024-08-06

Thank you all for your feedback. Please refer to the attached PDF for some of our main evaluation results as well as an additional runtime comparison: (1) Evaluation results on CommonGen (dev & test) for FUDGE, NADO, NeuroLogic A\*esque, GeLaTo and Ctrl-G. (2) Runtime analysis on CommonGen (dev) for NeuroLogic A\*esque, GeLaTo and Ctrl-G. (3) Evaluation results on Interactive Text Editing for TULU2-7B, GPT3.5, GPT4 and Ctrl-G. (4) Runtime analysis on Interactive Text Editing for TULU2-7B and Ctrl-G.

Upon your request, we have also released the output of Ctrl-G via an anonymous link. The link contains the output files as well as an html-based data visualization tool for you to browse the outputs. The link is sent to the AC following the instrutions for authors.

---

### Decision · Program_Chairs · 2024-09-25

**Decision:**

Accept (poster)

**Comment:**

The study proposes an inference-time method based on HMMs and finite state automata to constrain the output of an LLM. The study presents encouraging results in terms of human evaluation and quality of the generated output.

The problem is interesting and important. However, the contribution could be seen as somewhat incremental. The reviewers also voiced concerns about clarity and appropriate comparison with baselines. These concerns were largely addressed in the author/reviewer discussion.